# Assessing the supply for a basic urban service demand-with a focus on water-energy management in Addis Ababa city

**Bedassa Dessalegn Kitessa[1]\*, Semu Moges Ayalew[2], Geremew Sahilu Gebrie[1], Solomon T/mariam Teferi[1]**

**1** Addis Ababa Institute of Technology, Addis Ababa University, Addis Ababa, Ethiopia, **2** School of Civil and Environmental Engineering, University of Connecticut, Mansfield, Connecticut, United State of America

\* bedassa.dessalegn@aait.edu.et

**Data Availability Statement:** All relevant data are within the manuscript.

**Funding:** The author(s) received no specific funding for this work. Therefore, no funder has role

## Abstract

The demand for water-energy (WE) should be addressed with their sustainable supply in the long-term planning. The total energy demand was estimated to be around 14,000000 and 53,000000 MWh for 2030 and 2050 years respectively. These years' predicted water demand was 0.4 and 0.7 billion-cubic-meter. Based on the estimated energy and water demand, sustainable supply through WE management were determined. In 2030 and 2050 the water supply-demand balance index is around 1, showed water demand will be met for respective years, whereas the energy supply-balance after the intervention become around 0.9 and 0.7. The study results clearly predicted future WE demand of Addis Ababa city and have been put their quantified supply suggestion.

## 1. Introduction

Water-energy (WE) are the basic needs of society. The three basic socio-economic drives such as population, per capita income (PCI), gross domestic product (GDP) and technology affect highly the WE demand [1, 2]. City governors need to ensure sustainable water-energy supply to meet the demand through protecting available resources and adopting technologies through water-energy conservation and demand management strategies, because greater than 50% of the global population currently lives in cities [3]. The population (X1) is targeted to reach 9.3 billion [4, 5], globally by 2050. Populations of Ethiopian urbans are expected to triple by 2037 by a growth rate of 5% per year [6]. In 2015, the city GDP (X2) and PCI (X3) are about 4.3 billion US$ and 1,359 US$ respectively [7]. The growing population on the Addis Ababa city, high urbanization rates and higher affluence stimulating consumption of WE are basic trends driving the future development of impacts and city needs for WE supply. Studies on WE demand indicated that different countries, regions, or cities have different research methods and perspectives on the WE demand [8], and there exist connections between WE and such a relationship can be influenced by socio-economic factors. The WE demand is predicted based on socio-economic factors using a regression model [9]. The regression model is the most popular implemented numeric predictions and a stochastic approach for modeling the

in study design, data collection and analysis, decision to publish, or preparation of the manuscript.

**Competing interests:** The authors have declared that no competing interests exist.

relationship between a variable WE system (y) and one or more socio-economic variables denoted by x, using a WEKA tool x [10].

## 1.1. Water Conservation and Demand Management (WCDM)

Water is fundamental for life in terms of quantity and quality. Most available water sources are surface water or groundwater. Surface water considered an essential source for water supply in many improving countries [11].

The water supply system of the Addis Ababa city is characterized by a low output capacity, inadequate networks and high system losses, high water use or high water demand, urbanization, and high population [12]. The city water demand met is not more than 60% [13] and it is expected to address the estimated 36.5% leakage of water supply in the system as a way of ensuring that more potable water is made obtainable for the population [14].

Addis Ababa city has very limited resources of surface and groundwater which plays an important role in the support of domestic needs and failure of whichever would result in a crisis [15]. Hence, water supply is an important sustainable development, availability, adequacy of the city water supply will be improved to achieve a sustainable goal [16]. Water management is the issue for sustainable development to maximize economy, secure production and reduce environmental impacts [17]. However, the city water planners have given priority in developing new water resources to the supply side of water development but, demand side management and improvement of patterns of water use has received less attention [18]. The intervention in the water supply management focuses on expanding and diversifying a new water supply sources which include maximizing stormwater storage through distributed water harvesting structures and increasing existing water storage facilities [19].

WCDM is the implementation of strategies, policies; measures aimed at influencing water conservation and demand to achieve sustainable development use of scarce water resources [17]. The WCDM is any socially beneficial measure which reduces the consumptive use from surface or groundwater using strategies of efficient fixtures, water efficiency labelling, and conservation awareness programs [20, 21]. The norms and standards in respect of tariffs for water states that water tariffs must be cost reflective and must provide for a rising block tariff structure, which will not only support the viability and sustainability of water supply services to the poor but will also discourage wasteful or inefficient water use [22]. It has been shown all over the world that metering and billing creates strong incentives for consumers to use water sparingly [23]. The elasticity studies have shown a 1 to 3% reduction in demand for every 10% increase in the average monthly water bill [24]. The interpretation provides a more conservative result and suggests a 3% reduction for a 20% increase in the water tariff [25]. Water distribution networks for water supply having large leakage losses constitute a significant portion of water demand, and thereby leakage control is one the important measures to reduce water loss [26, 27]. The toilet and showerhead are most of the indoor water use, toilet account about 30% with averaging 5 flushes per capita per day, and replacing the efficient toilet with a low flow model will conserve water or implementing the efficient water appliance has a great impact on reducing water consumption [28].

Water demand management is any socially beneficial measure which reduces the consumptive use from surface or groundwater [20]. It is the development and implementation of strategies aimed at influencing demand, so as to achieve efficient and sustainable use of a scarce resource and widely considered as a promising path towards the sustainable development of water [16, 17].

In urban areas, relying on water distribution networks for water supply having large leakage losses constitutes a significant portion of water demand, and thereby leakage control is one the

important measures to reduce water loss [26]. Adopting the low-flow appliances (e.g., toilets, showerheads, and washing machines) can save water usage [27]. Water demand management strategies include efficient fixtures, water efficiency labelling, and conservation awareness programs [21].

The UN has recognized that African cities urgently need to develop and implement effective water conservation and water demand management strategies. The toilet account 30% of all indoor uses and is one of the most water uses, averaging 5 flushes per capita per day. An inefficient showerhead can use more than 20 liters per minute but a water efficient showerhead only uses about 8 liters per minute which can save significant amount of water without reducing the quality of service to the user [29]. Replacing an inefficient toilet with a low flow model will conserve water [28].

## 1.2. Energy Conservation and Demand Management (ECDM)

Addis Ababa city is a grid-connected electric energy system, which is accompanied by high energy demand and limited energy supply. To ensure reliable power supply security in the city, the government has a grid expansion plan [30]. Also, to enhance energy supply, the city has to increase the alternative sources of energy and concentrating more on energy efficiency which is the ultimate for achieving supply and demand balance [31].

Increasing the energy supply is essentially a strategy to reduce the risks that derive from energy use. In the USA, least-cost planning assessments of utilities forward plans to encourage adopting energy demand management (EDM) options [32]. This approach is more profitable for a utility to provide for the increasing demand for energy by investing in the management of demand than investing in developing a new supply capacity. Energy efficiency (EE) reduces the cost of generation, transmission, and distribution in the long-term plan and reduces the cost of service by 25% [33]. Lighting is essential for human activities (e.g., industrial, residential and commercial sectors) and it is desirable to have good, effective and efficient lighting. Existing inefficient light sources, retrofitting them with efficient ones offers an opportunity for sustainable energy.

In Ethiopia, the energy conservation by 2030 represents 53% of total energy consumption with a cumulative saving of 37,966 GWh over the coming 12 years [33]. The energy efficiency and conservation program includes standards and labelling, public sector energy efficiency, awareness, training and accreditation. In 46698 households of Ethiopia, the annual energy consumption for household lighting with CFL and incandescent bulbs lamp is estimated as 37 and 263 kWh respectively and the energy saving due to CFLs is 226 kWh [33]. Consequently, electric motor standards save energy demand in the industry by implementing IE-2 electric motors as MEPS for all energy-intense industries [33].

Saving water is also a basic important for energy sustainable development, and they are also interdependent and mutually reinforcing, this system is termed as water-energy nexus. The annual energy use by drinking water and wastewater treatment systems is about 39.2 and 30.2 billion kWh, constituting a combined 1.8% of all electricity used in the USA [34]. Consequently; according to AAWSA data, in Addis Ababa city the annual potable water and wastewater treatment use is about 0.6 billion and 6.4 million kWh respectively. The water saving potential of Addis Ababa is from 30 to 45%, this saves the energy potential of from 12 to 30% [35], indicating that saving water in the city will reduce the energy requirements.

This paper aims to predict the future water-energy demand (2016–2050) using the regression model and assesses sustainable water-energy supply to improve the future city demand through considering ECDM and WCDM based on predicted demand.

## 2. Methodology

### 2.1. Overview of Addis Ababa city

Addis Ababa city is located at 38˚44'E and 9˚1'N, as well as it is home to 25% of the city population in Ethiopia and one of the rapidly growing in Africa [36]. Addis Ababa's GDP is growing by 14% per year and has 50% of national GDP contribution [37]. The city water scarcity is become significant due to the high growth of urbanization and individual water demand. The total water sourced from groundwater and surface water is about 0.45 million $m^3$/day and 36.5% of the water is lost due to leakage [37].

### 2.2. Methodological framework

The WE demand prediction is based on developing a mathematical relationship between the predictor socio-economic variables with the dependent variable WE consumption based on regression model using the WEKA tool. Flow diagram in predicting WE demand is indicated in Fig 1.

The study that investigates the interactions between supply and demand of water-energy for designing strategies in support of the decision making is indicated in Fig 2.

### 2.3. WE demand prediction model

The regression equations were providing a fundamental explanation for an actual trend and high goodness of fit [38]. The multivariate linear regressions are commonly used in the water-energy demand prediction. The linear regression model can extend to a multivariate linear relationship as expressed in Eq (1).

$$y = f(x, \beta) = \beta_0 + \beta_1 x_1 + \beta_2 x_2 + \beta_3 x_3 + \cdots + \beta_p x_p \tag{1}$$

Where $\beta_1, \ldots, \beta_n$ and $\beta_o$ are coefficients of regression and constant respectively and $x_1, \ldots, x_n$ are explanatory variables (population, GDP and PCI) and whereas y (water-energy consumption) is the dependent variable.

The widely used model accuracy measures are Coefficient of determination ($R^2$), Mean Absolute Error (MAE) [39], Mean Absolute Error (MAE) and Root Mean Squared Error (RMSE).

### 2.4. WEKA tool

This tool is developed at the University of Waikato in New Zealand. It is a collection of machine learning algorithms for solving real-world data mining problems. A number of data

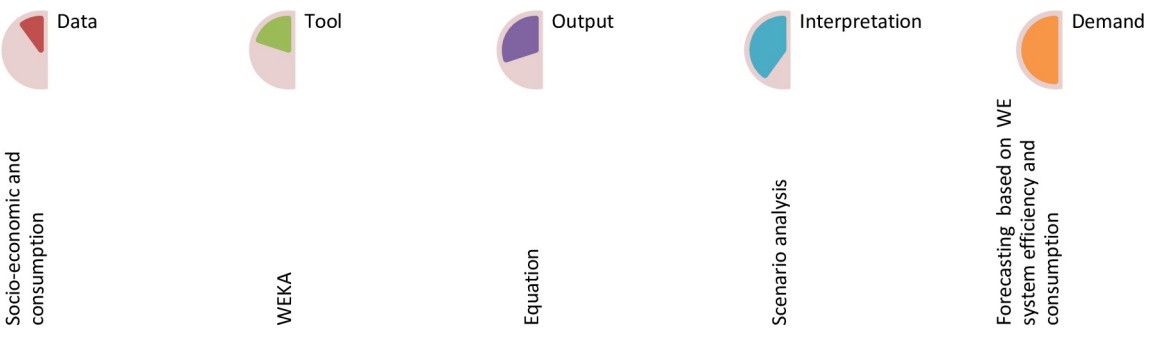

**Fig 1. Flow diagram to predict the water-energy demand.**

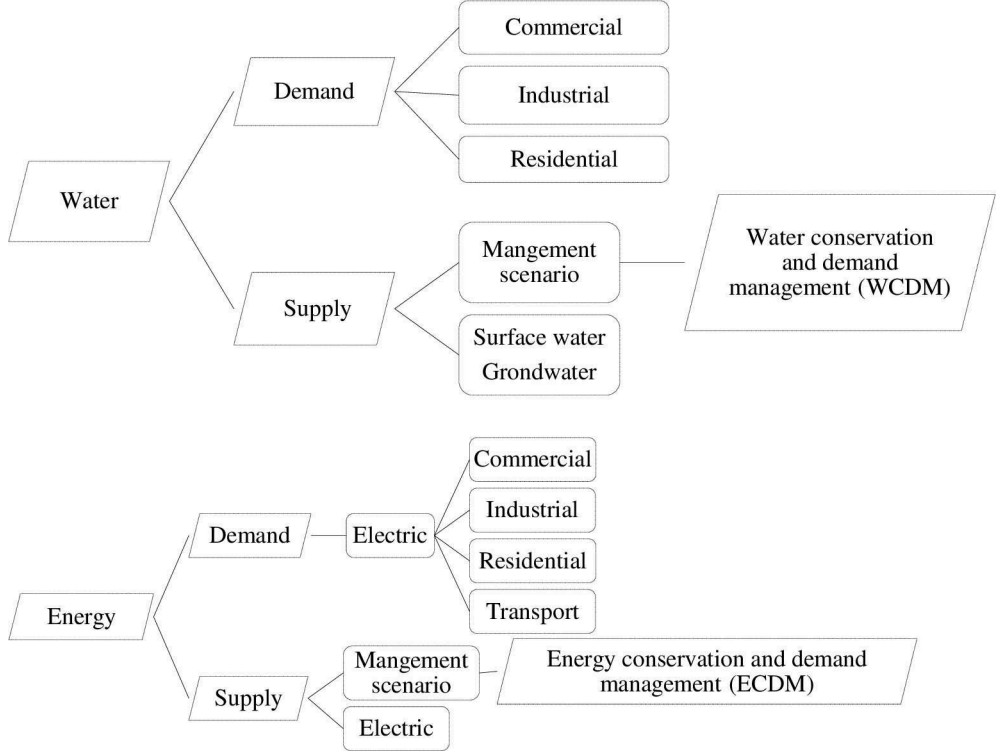

**Fig 2. Framework of water-energy supply and demand analysis.**

mining methods were implemented and experimented in the WEKA tool. Some of them were based on probability and regression was implemented. WEKA is the most data mining software over the others as it is open source and deployed on any problem [40].

## 2.5. Data used and analysis

Socio-economic parameters were collected from Bureau of Finance and Economic Development (BoFED) for GDP ($X_2$) and the Central Statistical Agency (CSA) for population ($X_1$). The consumption was collected from Addis Ababa Water and Sewerage Authority (AAWSA) for water and Ethiopian Electric Utility (EEU) for energy. See S1–S3 Tables for the socio-economic, water and energy consumption trend of Addis Ababa city respectively.

The scenarios for population growth rate are considered to predict the future populations which are indicated as follows:

Scenario 1(Business as usual): A population growth rate of about 3% is expected [41]. This is based on rural-urban migrants due to economic growth and migration of population to secondary cities.

Scenario 2 (High population growth rate): The urban population growth rate for Addis Ababa city is influenced by policies such as megaprojects. These various factors estimates come to a 5.2% [42]. Therefore, for each growth rate scenario, future population is estimated using Eq (2).

$$P_t = P_o(1 + GR) \tag{2}$$

Where $P_t$ is future population, $P_o$ is current population and $GR$ is population growth rate

The scenarios of GDP growth rate are analyzed as follows:

Scenario 1(Business as usual): During the Growth and Transformation Plan II (GTP II), the city GDP grew at about 18% annually, but to achieve this it requires high investment [43]. Scenario 2 (Lower GDP growth rate): The growth rate will continue by 11% per year after 2017 years [43]. Therefore, based on the existing GDP and growth rate (%), future GDP is estimated using Eq (3).

$$GDP_t = GDP_o(1 + GR) \tag{3}$$

Where $GDP_t$ is future GDP, $GDP_o$ is current GDP and $GR$ is growth rate

Although, per capita income (PCI) is main influencing parameter to predict the water-energy demand, considering the GDP and population; the PCI will be predicted using Eq (4).

$$PCI = \frac{GDP}{Population} \tag{4}$$

## 2.6. Existing and planned water supply by utility

The total water produced from existing surface and groundwater sources of the city is 574,000m$^3$/day at the end of 2018. However, there are interferences faced on the entire production potential of 574,000m$^3$/day due to underutilizing the maximum production capacity of sources due to frequent power interruptions, a high quantity of water loss. Due to these, on average, the amount of water supplied is only 460,000m$^3$/day. Proposed water supply projects besides the existing water supply sources, AAWSA has developed a long term business plan to meet the ever increasing demand of the city. The summary of existing and future production capacity of surface water and groundwater source is indicated in Table 1.

The total existing and planned surface water and groundwater supply of Addis Ababa city is summarized as indicated in Fig 3.

Table 1. Existing and future water supply capacity (m$^3$/day) from surface water and groundwater source.

| Surface water | Existing | Future | Remark |
|---|---|---|---|
| Legadadi and Dire dam | 170,000 | 195,000 | Add 25,000m$^3$/day by upsizing the existing capacity |
| Gefersa | 30,000 | 30,000 | |
| Gerbi | - | 73,000 | Medium-term plan |
| Siblu | - | 600,000 | Long-term plan |
| Total | 200,000 | 898,000 | |
| Groundwater | Existing | Future | Remark |
| Akaki Phase I | 12,000 | 12,000 | |
| Akaki Phase II | 70,000 | 70,000 | |
| Akaki Phase IIIA | 47,000 | 47,000 | |
| Akaki Phase IIIB | 70,000 | 70,000 | |
| Legadadi Deep Well Phase I | 40,000 | 40,000 | |
| Springs, boreholes and wells | 91,000 | 94,000 | Add 3,000m$^3$/day from pocket areas: short-term plan |
| Koye Feche | 44,000 | 50,000 | Add 6,000m$^3$/day from Klinto area: short-term plan |
| Legadadi Deep Well Phase II | - | 86,000 | Medium-term plan |
| Ayat North Fanta | - | 68,000 | Medium-term plan |
| Sebeta-Holeta | - | 100,000 | Medium-term plan |
| Total | 374,000 | 637,000 | |
| Total (Surface and Groundwater) | 574,000 | 1,535,000 | |

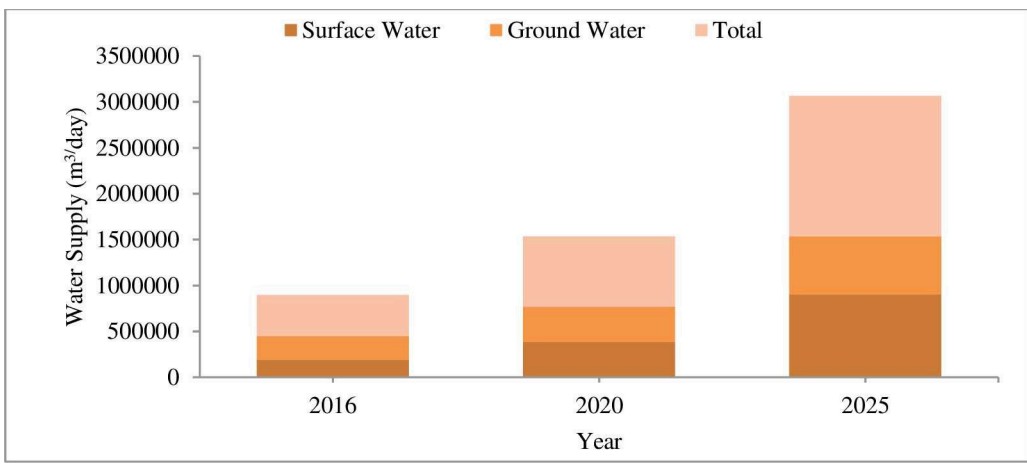

**Fig 3. Water supply capacity from surface and groundwater source [44].** The total existing and planned water supply of Addis Ababa in 2025 will be 0.56 BCM (1534247 m³/day) and this value is expected to reach 2030 and 2050 when there is no addition water supply scheme.

## 2.7. Existing and planned energy supply by utility

The Ethiopian Electric Power Corporation (EEPCo) is the major supplier of electricity supplemented by Addis Ababa city. The city power supply system is the interconnected system (ICS) which has grid connections and is mainly supplied from hydropower plants. A load of Addis Center (ADC) and Addis North (ADN) substations is 64 and 47 MW respectively. A load of ADC substation in 2017 was 64 MW and with a growth rate of 7.8% and a power factor of 0.95, the load will exceed the available total capacity of around 190 MW of the substation in the year 2032 [45]. The load of ADN substation in 2017 was 47 MW and with a growth rate of 7.8% and power factor of 0.95, the load will exceed the available total capacity of around 71 MW of ADN substation in the year 2022 [45]. The load of Addis Eastern or Weregenu substation in 2017 was 54 MW. Based on the growth rate of 7.8% and the power factor of 0.95, the load will exceed the installed capacity of 119 MW substations in the year 2028. Existing and planned power supply from sub-stations for Addis Ababa city is indicated in Fig 4.

## 2.8. Water supply interventions

**2.8.1. Source substitution by storm water storage (rainwater-runoff harvesting).** To explore the stormwater potential, three basic variables are considered as mean annual rainfall, catchment area and rainfall run-off coefficient. According to the rainfall measured by National Metrological Agency (NMA) at five meteorological stations (Entoto, Akaki, Bole, Kotobe and Ayertena), the average annual minimum and maximum rainfall are 800 and 1300mm respectively as indicated in Fig 5. The minimum standard requirement of rainfall for the rainwater harvesting system is 300 mm per year [46] and the minimum mean rainfall value of Addis Ababa is 845mm, which is significantly higher than the standard value and the requirement is satisfied. The average rainfall of about 1115 mm (2005–2015) is used for the calculation of the stormwater potential of Addis Ababa city.

The catchment area of Addis Ababa is about 540 km² which is considered as the land area where the surface water from precipitation sources is collected and drained towards a common exit. A runoff coefficient based on the surface feature is assessed as for local streets ($c_1 = 0.5$), for parking lots ($c_2 = 0.4$) and for green space ($c_3 = 0.2$) [47]. It has been observed that most of Addis Ababa's buildings used a Galvanized Iron sheet of roofing material for which its runoff

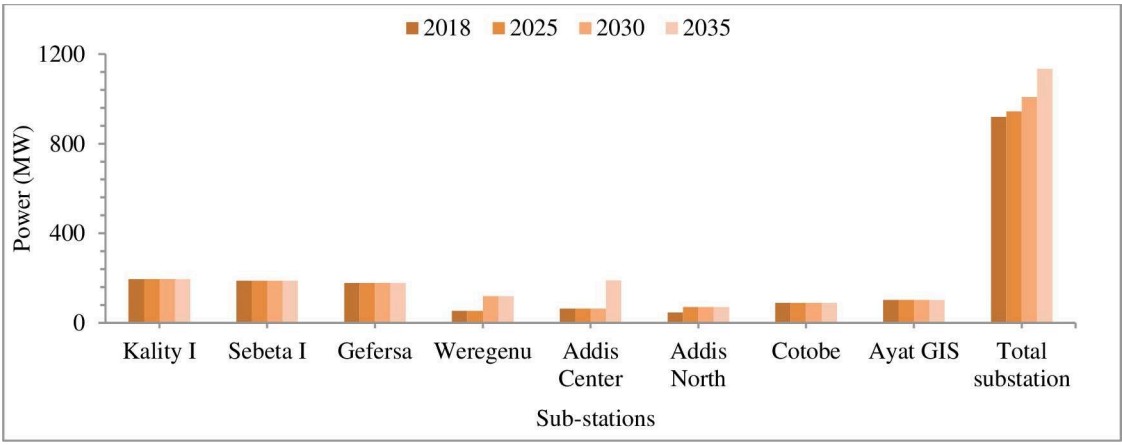

**Fig 4. Power load of Addis Ababa city sub-stations.** The total energy supply of sub-stations in 2030 and 2050 will be 8836 GWh (31 PJ) and 9943 GWh (36 PJ) respectively. For the respective years, the total power supply of Addis Ababa city are 1009 and 1135 MW respectively.

coefficient ($c_4 = 0.9$) [48]. The average runoff coefficient is taken as 0.5. The potential of rainwater runoff can be calculated by using Eq (5) [49].

$$Q = C_{average} \times A \times R \qquad (5)$$

Where; Q is rainwater runoff in $m^3$, R is the average annual rainfall (mm); A is catchment area ($m^2$) and $C_{average}$ is average runoff coefficient.

**2.8.2. Distribution system water loss reduction.** The water loss analysis for the annual NRW level in Addis Ababa is 38.2 MCM (40% system input volume), from the total, the physical losses are 27%, the commercial or apparent losses is about 11% and unbilled authorized consumption 2% [28]. Furthermore, economical leakage reduction has to be about 10% of the water losses is an economical leakage level or 30% of savings [28]. The attention of AAWSA is to reduce NRW using district area metering to investigate invisible leakage in the water distribution service network to 20% which is true in many countries [50]. Water use efficiency (leakage improvement and other growth of technology for efficient water supply) is improved

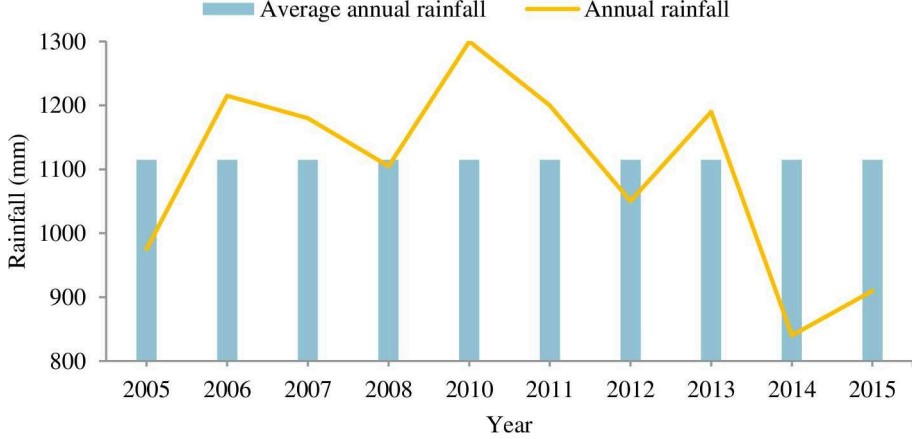

**Fig 5. Variations of average annual rainfall.**

by 2% for the reference year (2012); and annual improvement in efficiency is expected to increase by 5% until 2039 [51].

**2.8.3. Replacing older inefficient toilets fixture.** In the home built in the 1980s, toilets are designed to flush with 13 liters, replacing an inefficient toilet with a low flow model will conserve water. The Addis Ababa Water and Sewerage Authority (AAWSA) have been implementing a program targeting the replacement of older plumbing fixtures. A low flow fixture is defined according to plumbing fixture standards of double flush toilet 6/3 or 4 liters per minute.

**2.8.4. Install low-flow shower heads and faucet aerators to residential users.** A low-flow retrofit for showerheads, faucet aerators and other applicable retrofit items are a package of water saving devices that can assist residents to save water at home. The distribution of low-flow (high efficient) retrofit to the residents can reduce a number of less efficient retrofit. A low-flow showerhead that is highly efficient uses 7.6 liters/minute. The efficient kitchen and low-flow lavatory faucet aerator provide an even spray pattern at 7.6 and 3.8 liter/minute respectively.

**2.8.5. Install low-flow toilets and urinals in government and business buildings.** AAWSA should demonstrate water conservation and demand management through replacing inefficient fixtures with high efficiency fixtures in government and business buildings not only conserve water for the local government, it provides an opportunity for public awareness and education. High efficiency replacement fixtures include high efficient toilet (HET), 6/3 liter per flush double flush or less, and high-efficiency urinals, 3.8 liters per flush or less.

**2.8.6. Conservation and improving billing system.** Water tariff ensures sustainable and equitable development, as well as efficient and effective management. It should be realistic and generate adequate revenue to fund operations, capital expenditure and repayment of loans and should be structured to include, amongst others, the following: Recovery of the cost of water purchases; recovery of overheads including operational and maintenance costs and provision for the replacement, upgrading and expansion of water services. According to AAWSA, the charge of 0.15 USD per month was levied for the indirect financial situations (indigent tariff); while an intermediate rate of 0.6 USD per month was charged for sub-economic households from 2025–2050 [28]. Improving billing system by replacing the existing the billing software is used to consider certain functionality to facilitate conservation.

**2.8.7. Cloth washer rebate.** The AAWSA water utilities use water for cloth wash and efficient cloth wash is an important opportunity to attain substantial energy savings and simultaneously water saving. Efficient water fixture of cloth washer water saves energy of 1213 kWh with 7.2 m$^3$ of water saving.

The interpretation value for water conservation and demand management option with their percent of saving will be expected to increase annually (2010–2050) as indicated in Table 2.

As indicated in Table 2, the water saving potential by using WCDM measure for a baseline period (2010) was mentioned in million liters per day (MLD) [52].

## 2.9. Energy supply interventions

**2.9.1. Renewable energy (solar energy).** The technical potential is derived based on constraints regarding the area that is convincingly available for energy generation [53].The theoretical potential is the amount of annual solar radiation in suitable areas for solar applications, which should take into account different constraints in the assessment phase to achieve the appropriate areas [53]. Assessing technical potential of solar PV energy is taken in to consideration using a diagram flow indicated in Fig 6.

**Table 2. Percent growth of water saving by conservation and demand management option [28].**

| Measures | Saving in base period (MLD) (2010) | % of demand saving in base period (2010–2016) | % of demand saving (2040) | % growth rate of saving (2010–2050) |
|---|---|---|---|---|
| Replacing older inefficient toilet fixtures | 5 | 1 | 5 | 4 |
| Water loss reduction | 25 | 5 | 7 | 0.4 |
| Conservation and improving billing system | 139 | 28 | 46 | 0.6 |
| Distributing low flow showerhead and faucets | 2 | 0.4 | 0.7 | 0.7 |
| Installing low flow toilets and urinals | 15 | 3 | 5 | 0.6 |
| Cloth washer rebate | 3 | 0.6 | 0.9 | 0.5 |
| Total | 189 | 38 | 65 | 0.7 |

Addis Ababa's average daily solar radiation and bright sunshine hour of 4.99 kWh/m$^2$/day and 7.3 hr are observed respectively from national metrological agency (NMA) and NASA for a year (2013–2016). The daily solar radiation and hourly sunshine of Addis Ababa city is shown in Figs 7 and 8 respectively.

The total roof area of the city is found to be 5,310 ha [48]. Different studies account in computation of rooftop space, pitched roof orientation, percentage of flat rooftops and the size of the solar panels. In computation formulas, $f_o$ represents the fraction of buildings with properly oriented roof area for rooftop solar, $r_{flat}$ represent proportion of flat rooftops which is equaled to 5% of total rooftop area of buildings and conversely, $r_{pitch}$ is the proportion of pitched rooftop buildings which is equal to 95% [54].

Considering all flat rooftops buildings are unaffected by their rooftop orientation. This means, they have a flat rooftop and all the direction of the building is facing to receive optimal sun exposure, which is indicating that the reduction factor for flat rooftops ($f_{flat}$) is equal to 100%. For pitched rooftops, the reduction factor for pitch rooftop ($f_{Pitch}$) value is equal to 50%. Taking these values, the optimal orientation for rooftop solar is calculated as Eq (6) [54].

$$f_0 = f_{flat} \times r_{flat} + f_{pitch} \times r_{pitch} \tag{6}$$

Similarly, a fraction of available rooftop area suitable ($f_s$) for solar panels when accounting for unknowns such as shadow and area needed for installation is taken as 0.3 [55]. The suitable

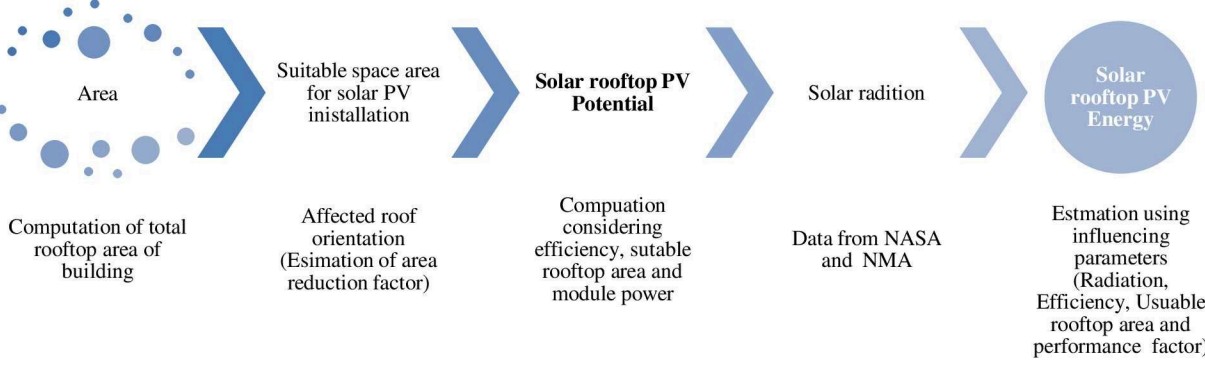

**Fig 6. Schematic of methodology PV potential assessment.**

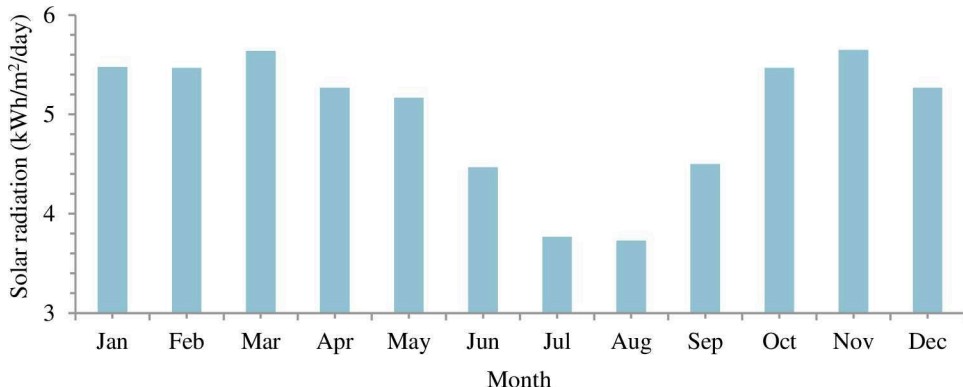

**Fig 7. Daily global solar radiations.**

rooftop area space for solar PV installation is computed from Eq (7).

$$A_{PV} = A_{total} \times f_0 \times f_s \tag{7}$$

Where: $A_{PV}$ is suitable rooftop area for solar, $A_{total}$ total rooftop area and $f_0$ and $f_s$ are reduction factor

The parameters of $r_{flat}$, $r_{pitch}$, $r_{pitch}$ and $f_s$ of around 5, 95, 50 and 16% are respectively taken in to consideration for the case of Addis Ababa city to estimate the suitable rooftop area for solar PV installation. Then, around 8.3% of available rooftop area (5310ha) is suitable area space (around 442 ha) for installation of PV.

The rooftop PV technical potential for rooftop building has been estimated as the direct power rating (kWp) under Standard Test Conditions (STC) from Eq (8) [56], as well as for energy is given in Eq (9).

$$TP = A_{PV} \times P \times \eta \tag{8}$$

$$\mathbf{E_{Pv} = TP \times t} \tag{9}$$

Where TP is technical potential of solar PV (kW), p is power produced by the module per a unit area (kw/m$^2$), $A_{PV}$ is the usable roof area of urban building (m$^2$) and $\eta$ the module efficiency (%), $E_{pv}$ is energy potential of solar PV (kWh), t is bright sun shine hours (hr/day)

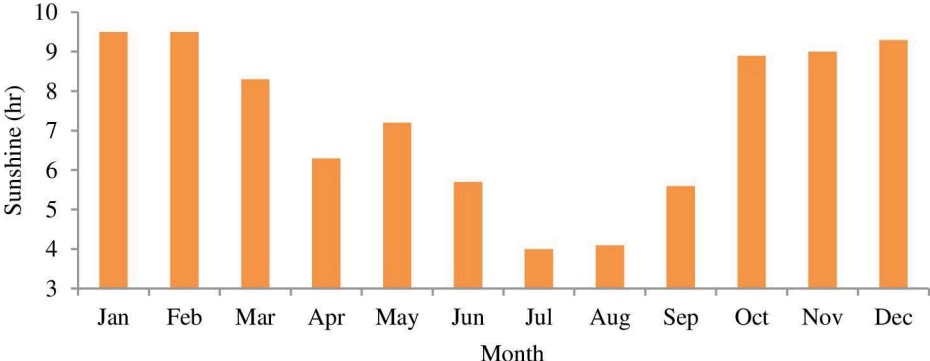

**Fig 8. Hourly average bright sunshine hours.**

A Solar World module is taken as reference solar module and its detail characteristics are presented in Table 3.

**2.9.2. Distribution system energy loss reduction.** Electric energy is lost as it flows from generation to end-use with the majority of the losses attributed to the distribution system. The national (Ethiopia) energy loss is about 18% of the electric energy generated [58]; this showing an efficiency of the system is 82%. Reducing the system loss from 18% to 12% as a first stage has able to save about 48 MW which is 6% of the total electric power (800 MW) [59]. The energy saving is also 6% of the total yearly energy generated (3,570 GWh) which is equal to 214.2 GWh. According to AADMP, the energy distribution loss in Addis Ababa city is 19% in 2016 and expected to decreases to 14.4% in 2019 [45]. The AADMP planned to improve distribution loss to 9% in 2034 and it reaches around 6% in 2050. Therefore, taking 19% of energy loss as a baseline, loss intervention is considered as it will decrease to 6.65% in 2050. Reducing the system loss in 2050 from a baseline of 19% to 6% has able to save about 13% of the total yearly electric energy supply and the analysis of the energy is given using Eq (10) and (11).

$$EL = \frac{19}{100} \times TES \tag{10}$$

$$ELS = ELR - EL \tag{11}$$

Where EL is energy loss, TES is total energy supply, ELS is the energy loss saving, and ELR is the energy loss reduction.

**2.9.3. Replacing efficient energy lighting.** Most of the energy is used for the provision of lighting in both residential and commercial buildings. In Addis Ababa city, the lighting consumption account for around 40%, 50% and 30% of commercial, residential and industrial energy consumption respectively. The efficacy of incandescent lamp lighting varies from 10 to 25 Lumens/watt while the Compact Florescent Lamp (CFL) is around 75 Lumens/watt, three times that of the incandescent lamp. Two-third of electric energy is saved for lighting by replacing the incandescent lamps with CFL [33]. In this study, two-third of electric energy is saved due to lighting by replacing with CFL [33] for residential, industrial and commercial sectors and these sectors replacing CFL will be expected to increase annually by 4% (2020–2050). The Eq (12), (13), (14) and (15) are used for the computations of energy by incident lamp lighting (ILL) and energy saving by CFL.

$$ILL = \frac{4}{10} \times CEC \tag{12}$$

$$ILL = \frac{5}{10} \times REC \tag{13}$$

$$ILL = \frac{3}{10} \times IEC \tag{14}$$

$$SCFL = \frac{2}{3} \times ILL \tag{15}$$

**Table 3. Characterization of reference module [57].**

| Manufacturer | SolarWorld |
|---|---|
| Module type | Module type |
| Model | Sunmodule Plus SW 260 poly |
| Dimensions | Length: 1675 mm, Width: 1001 mm, Thickness: 31.0 mm, Area: 1.67 m$^2$ |
| Nom. Power (at STC) | 260 Wp |
| Efficiency | 15.5% |

Where ILL is incident lamp lighting, CEC is commercial energy consumption, REC is residential energy consumption, IEC is industrial energy consumption and SCFL is saving by a compact fluorescent lamp.

**2.9.4. Replacing efficient industrial electric energy machines.** In many industrial plants, motors systems consume over 50% of the total electricity used in the industry. Industrial electric machine (IEM) standards save energy demand in the industry by implementing IE-2 electric motors as MEPS for all energy-intense industries [33]. About 70% of industrial energy is ended by electric machine (EM) in Addis Ababa city and about 60% of energy consumption is expected by replacing efficient machine (EM). Replacing the efficient machine will grow annually by 4% (2020–2050). The energy use by the machine is given using Eqs (16) and (17).

$$IEM = \frac{7}{10} \times IEC \quad (16)$$

$$EM = \frac{6}{10} \times IEM \quad (17)$$

Where IEM is an industrial electric machine, IEC is industrial energy consumption; EM is efficient machine.

**2.9.5. Replacing efficient electric mittad.** In Ethiopia, an important electric energy load in residential use is the electric energy Mittad which consumes on average of around 3 kW. Some studies demonstrate that the efficiency of baking can be improved by more than double using induction heating rather than resistor heating [52]. For 300,000 number of mittad, and operate for 2hr per week the total annual energy consumption become (3 kW x 52 weeks x 2 hours/week x 300,000 mittads) 93.6 GWh [52]. For the electric energy consumption is halved due to the induction heating (IH) rather than the resistor heating (RH) the energy saving becomes 46.8 GWh. The energy consumption is taken as a maximum of about 4 days per week for Addis Ababa city as well as number mittad is seemed with the household. The energy use analysis is given using Eqs (18) and (19).

$$RH = N \times PRH \times t \quad (18)$$

$$SIH = \frac{1}{2} \times N \times PRH \times t \quad (19)$$

Where SIH is saving by induction heating (GWh), RH is resistor heating (GWh), N is number of urban households using induction heating, PRH is the power of resistor heating (kW), and t is operational time (year).

**2.9.6. Energy efficient electric stove.** The most common stove is 22 cm (single and double stove); the capacity of the single stove (SS) and double stove (DS) is 1.4 and 2.8 kW respectively with the usage of 3.5 hr per day. Tests made on the existing electric stove of the 22 cm diameter (the most common stove size) show that EE of stoves could be improved by at least 24% [60]. This makes a power demand reduction of 0.4 KW per stove. For the commonly used single cook top 22 cm diameter stove, there could be up to 0.4 KW x 1 stove x 3.5hrs/day x 365 days = 511 KWh energy saving per year per consumer and whereas for the double cook top type the saving will be 1,022 KWh. The energy uses of these stoves are computed using Eq

**Table 4. Annual percent of replacing the efficient energy saving.**

| ECDM Measures | Baseline % (2016) | % growth (2016–2030) | % growth (2030–2045) | All achieved (2045–2050) |
|---|---|---|---|---|
| Energy loss reduction | 20 | 4 | 6 | All |
| Efficient electric machine | 20 | 4 | 6 | All |
| Electric lighting | 20 | 4 | 6 | All |
| Electric mittad | 20 | 10 | All achieved | All |
| Electric stove | 20 | 10 | All achieved | All |

(20), (21), (22) and (23) are used for computation.

$$SS = N \times PSS \times t \tag{20}$$

$$DS = N \times PDS \times t \tag{21}$$

$$ISS = \frac{24}{100} \times N \times PSS \times t \tag{22}$$

$$IDS = \frac{24}{100} \times N \times PDS \times t \tag{23}$$

Where SS and DS are single and double stove (GWh) respectively, PDS and PSS are power (kW) of double and single stove respectively, N is number of urban households using single and double stove, ISS and IDS are energy saving by improved single and double stove respectively and t is operational time (year).

**2.9.7. Efficient energy water machine.** AAWSA can save the water potential of 30–45%; this can also save the electric energy potential of Addis Ababa city from 12 to 31% [35]. Electric energy conservation or demand management intervention is given in Table 4, considering the 25% saving for a baseline [60].

## 3. Results and discussion

### 3.1. Future socio-economic trend

As GDP and population are main drivers for predicting the WE demand, the projected socio-economic drivers are analyzed under this section. The projected population for the two scenarios is shown in Fig 9.

From the two scenario of future population, rapid population growth rate is taken for WE demand prediction. Hence, due to the mega-city project is continuing, the estimate of city population growth rate of 5% [42] is used for demand prediction. Consequently, from the two scenarios the city GDP grown at an annual average of 11% [43] is used in demand prediction, because it may be challenging to achieve the city GDP under 18% growth rates. The GDP projection under the two scenarios of GDP growth rate is indicated in Fig 10.

### 3.2. Predicted future water-energy demand

The variables and the final governing multivariate linear regression equations obtained by the WEKA model to predict the WE demand are given in S1 Appendix.

**3.2.1. Predicted water consumption.** The forecast involved applying the relationships between historical patterns and different scenarios concerning economic and demographic changes. Since the prediction are directly linked to predictor variables ($X_2$ and $X_1$) and other variables ($X_1$ and $X_3$) or ($X_2$ and $X_3$). The forecasted water values got to be understood within

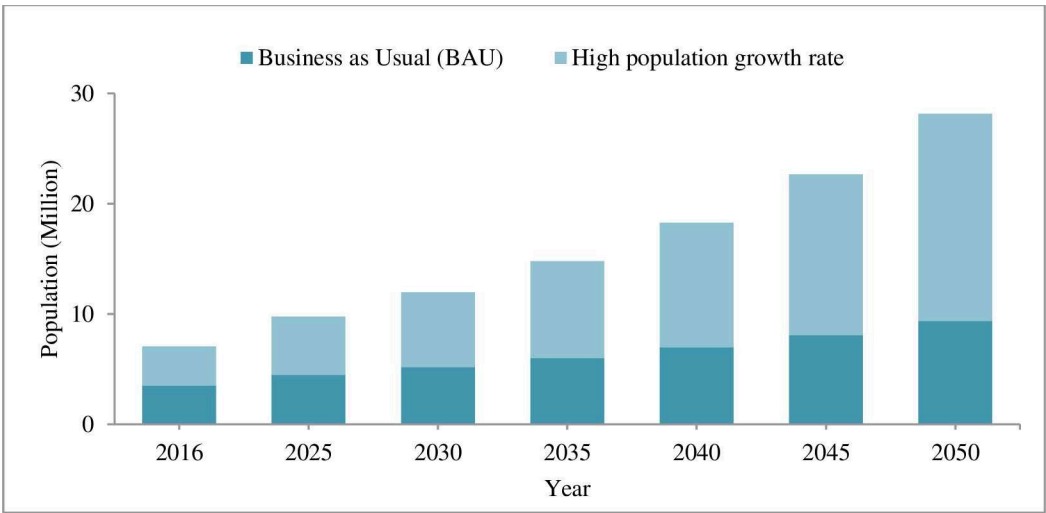

**Fig 9. Addis Ababa city projected population.**

the perspective of those predictor variables. If a specific predictor variables or scenario was compiled using $X_2$ and $X_3$ growth figures that were very far-away from the particular growth of actual patterns, and then the water forecasts generated for this scenario can also be unrealistically high. While scenario thinking can support planning and forecasting well, there are certain pitfalls to avoid when generating predictor variables [61]. The evaluation of variables was determined as given in Table 5.

The predicted water consumption in Million Cubic Meter (MCM) for the end-users is shown in Table 6.

**3.2.2. Predicted water demand.** In addition to water consumption, the loss in distribution system affects the water demand. Having a water loss reduction to 23% and 22% by 2030 and 2050 respectively [62], the water demand is estimated. Therefore, considering scenario 1

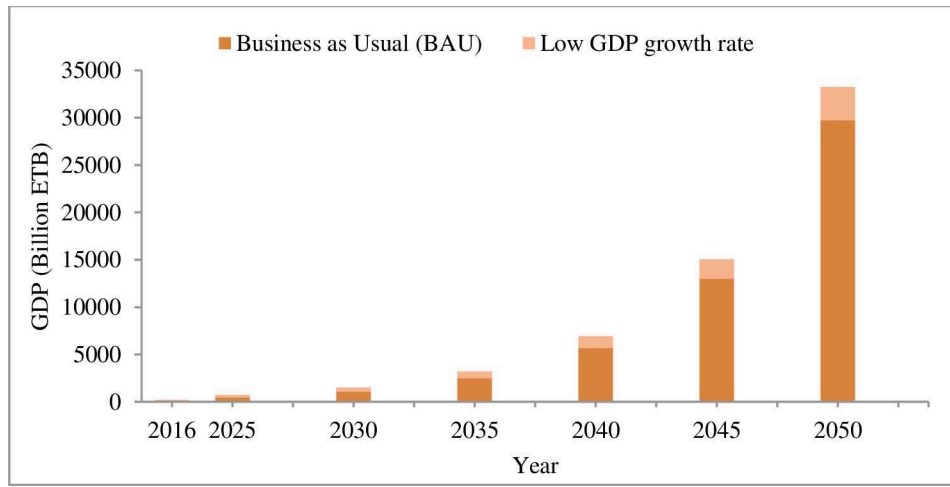

**Fig 10. Projected GDP of Addis Ababa city.** The GDP under business as usual is greater by 40% and 12% in 2030 and 2050 respectively as compared to the low GDP growth rate, this indicates a high gap between the GDP predicted between two scenarios.

**Table 5. Evaluation of drivers for water consumption prediction.**

| Sectors | Scenario | Drivers | Parameters | | | | |
|---|---|---|---|---|---|---|---|
| | | | $R^2$ | MAE | RMSE | RAE (%) | RRSE (%) |
| Commercial | 1 | $X_1, X_3$ | 0.97 | 0.002 | 0.002 | 24 | 23 |
| | 2 | $X_2, X_3$ | 0.96 | 0.002 | 0.003 | 27 | 26 |
| Industrial | 1 | $X_1, X_3$ | 0.95 | 0.002 | 0.002 | 34 | 31 |
| | 2 | $X_2, X_3$ | 0.95 | 0.002 | 0.003 | 33 | 32 |
| Residential | 1 | $X_1, X_3$ | 0.98 | 0.003 | 0.003 | 8 | 8 |
| | 2 | $X_2, X_3$ | 0.98 | 0.003 | 0.004 | 8 | 9 |

($X_1$ and $X_2$) based water consumption for commercial and residential as well as, scenario 2 ($X_2$ and $X_3$) for industrial, the water demand is indicated in Table 7.

For the rapid population growth rate, the total water demand will be 679 MCM which has is insignificant gap compared to 686 MCM in 2039 [63]. The Addis Ababa city water demand is expected to reach 431 and 1199 MCM by 2030 and 2050 respectively. The domestic water demand will be 116 and 119 Liter per capita per day (LPCD) by 2030 and 2050 respectively, which are greater than 110 LPCD of the WHO recommend. The long-term plan of Ethiopia is to become a middle-income country with increase in income, designating domestic water demand would continue to upsurge. The growth rate of water demand in each sector is indicated in Fig 11.

**3.3.3. Estimated energy consumption.** In the transport sector, energy use was estimated for scenario 1: ($X_1$ and $X_3$), scenario 2 ($X_3$) and scenario 3 ($X_1$). Scenarios 2 and 3 are less accurate to the observed consumption for street-lighting or transport sector compared to scenario 1. Therefore, the consumption estimated based on $X_1$ and $X_3$ scenario was considered for demand prediction. In the case of commercial, residential and industrial sectors, the energy use was predicted based on scenario 1 ($X_1$), scenario 2 ($X_3$) and scenario 3 (Average). Scenario 1 ($X_1$) and scenario 2 ($X_3$) were gives nearly the same value of consumption. For exactness, the averages of two scenarios predict energy consumption for the three sectors. Evaluation of predictor variable used in estimation of energy consumption is indicated in Table 8.

The two scenarios ($X_1$ and $X_3$) were estimated approximately the same value of energy consumption in the industrial, commercial, transport and residential sectors. The average of two scenarios represents the actual consumption in these sectors; hence considering more drivers in consumption estimation increases the reliability of predicted consumption. Indicating that energy consumption is highly related to predictor variable $X_1$ and $X_3$. The predicted energy consumption is given in Table 9

**3.3.4. Estimated energy demand.** The total energy supply of Addis Ababa city was affected by loss in which about 15% of the total energy supply is lost by distribution system in

**Table 6. Water consumption based on drivers.**

| Sectors | Scenarios | Year | | | | | | |
|---|---|---|---|---|---|---|---|---|
| | | 2016 | 2025 | 2030 | 2035 | 2040 | 2045 | 2050 |
| Residential | 1 | 110 | 178 | 233 | 305 | 399 | 522 | 683 |
| | 2 | 116 | 179 | 231 | 299 | 391 | 511 | 672 |
| Commercial | 1 | 30 | 48 | 63 | 83 | 110 | 144 | 189 |
| | 2 | 30 | 44 | 58 | 77 | 104 | 449 | 206 |
| Industrial | 1 | 27 | 39 | 49 | 63 | 80 | 102 | 132 |
| | 2 | 27 | 43 | 57 | 67 | 83 | 104 | 128 |

**Table 7. Predicted the water demand (MCM).**

| Sectors | Year | | | | | | |
|---|---|---|---|---|---|---|---|
| | 2016 | 2025 | 2030 | 2035 | 2040 | 2045 | 2050 |
| Residential | 165 | 235 | 287 | 376 | 482 | 625 | 818 |
| Commercial | 44 | 63 | 78 | 103 | 133 | 173 | 227 |
| Industrial | 45 | 57 | 66 | 82 | 101 | 124 | 154 |
| Total | 254 | 355 | 431 | 561 | 716 | 922 | 1199 |

2019 [45]. As planned by Addis Ababa Distribution Master Plan, energy loss in power distribution network will decrease to 9 and 7% by 2034 and 2050 respectively. The future energy demands considering the loss for different sectors are indicated in Table 10.

The electric energy demand growth rate for different end-users (sectors) is indicated in Fig 12.

## 3.4. Analysis of water supply intervention

AAWSA water supply capacity (existing and planned) could not meet the overall demand from 2030 to 2050. Considering its limited water supply in Addis Ababa city, AAWSA will have to devote itself to improving its water availability to mitigate the dilemmas between supply and demand. The reason lies behind this might be that all the demand sites rely on surface water and groundwater as their only water source. The water management practices such as water conservation or demand (WCDM) and supply management option will propose as the future scenario to decrease and manage the unmet water demand of the city. Analysis of WCDM scenario indicates that with the effective implementation of different management strategies at the conservation and demand side (Efficient cloth wash rebate, providing efficient water loss reduction, replacing efficient retrofit and conserving water pricing) unmet water demand will grow to 328 MCM by 2030 and 532 MCM by 2050; this value is about 83% decrease when compared with the existing and planned water scenario (Reference scenario). In 2050, the expected possible percent of the water that will conserve from the total demand is indicated in Table 11.

The conservation and billing system will conserve more percent of water as compared to the other WCDM measures due to its less present cost value in water saving [28]. In the water supply management (WSM) scenario (developing stormwater storage), the annual stormwater

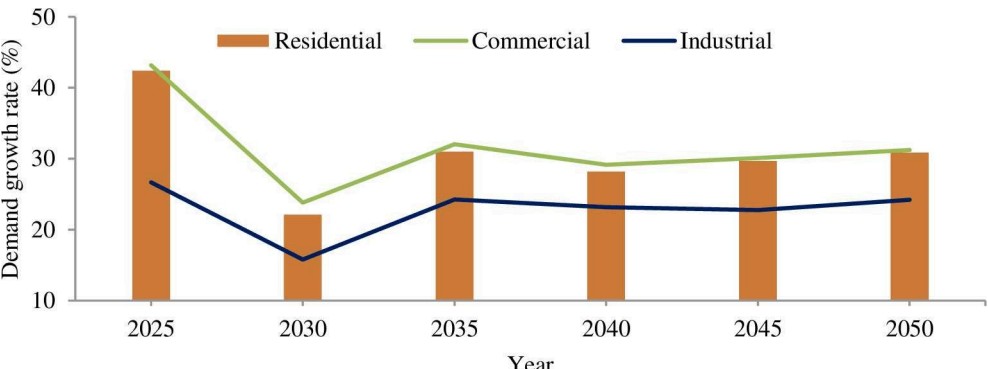

**Fig 11. Estimated water demand growth in the sector.** Residential sector has high water demand growth rate as compared to commercial and industrial sector. The demand growth rate for commercial, industrial and residential will be increases by 5%, 3.7% and 4.8% respectively annually from 2016 to 2050.

**Table 8. Evaluation of scenarios used in estimation of energy consumption.**

| Sector | Scenario | Drivers | Parameter | | | | |
|---|---|---|---|---|---|---|---|
| | | | $R^2$ | MAE (PJ) | RMSE (PJ) | RAE (%) | RRSE (%) |
| Transport | 1 | $X_1$ | 0.98 | 0.002 | 0.002 | 19 | 18 |
| | 2 | $X_3$ | 0.92 | 0.004 | 0.004 | 41 | 40 |
| Commercial | 1 | $X_1$ | 0.98 | 0.08 | 0.09 | 14 | 15 |
| | 2 | $X_3$ | 0.98 | 0.08 | 0.09 | 14 | 16 |
| Residential | 1 | $X_1$ | 0.95 | 0.2 | 0.3 | 26 | 31 |
| | 2 | $X_3$ | 0.95 | 0.2 | 0.3 | 26 | 31 |
| Industrial | 1 | $X_1$ | 0.96 | 0.2 | 0.2 | 24 | 28 |
| | 2 | $X_3$ | 0.95 | 0.2 | 0.3 | 26 | 31 |

potential or rainfall-runoff water (RRW) of Addis Ababa city is expected to be about 315 MCM and considering 50 to 60% of the stormwater potential can be harvested (stored) annually and expected as rainfall-runoff water harvesting (RRWH) of about 158 to 200 MCM. Consequently; considering about 200 MCM, the analysis of this scenario shows that unmet water demand will reach 271 MCM by 2030 and 998 MCM by 2050, which represents about a 64% increase in unmet water demand with respect to the Reference (Ref) case. Similarly, analysis of this water management approach suggests that unmet water demand will reach 332 MCM by 2050 or a 52% reduction in comparison to the Ref scenario. From the foregoing results, it is clear that a water management approach is needed to reduce the unmet water demand of the city. From the water conservation and demand management scenario in 2050, 54% of water savings from the demand is reflected.

Thus, the unmet water demand of the city will be minimized most in the water management scenario (WMS) and this will support the achievement of the Sustainable Development Goal (SDG6). Other studies showed that integrated water management strategies save more water than individual management approaches. Studies stated that structural water saving and advanced or technical water-saving technology scenarios can result in water saving potentials of 6.97% and 9.82% by 2050, respectively [64].

Fig 13, shows the surplus and deficit of water for the water management scenario (WCDM and WSM) and reference scenario. The negative and positive sign shows the water deficit and surplus respectively in each scenario.

**Table 9. Energy consumption ($10^3$ GWh) for sectors.**

| Sector | Drivers | Year | | | | | | |
|---|---|---|---|---|---|---|---|---|
| | | 2016 | 2025 | 2030 | 2035 | 2040 | 2045 | 2050 |
| Commercial | 1 | 0.7 | 2.3 | 3.6 | 5.3 | 7.4 | 10.2 | 13.7 |
| | 2 | 0.7 | 2.4 | 3.7 | 5.6 | 7.7 | 10.7 | 14.6 |
| | 3 | 0.7 | 2.4 | 3.7 | 5.4 | 7.6 | 10.5 | 14.2 |
| Industrial | 1 | 1.2 | 3.4 | 5.1 | 7.4 | 10.3 | 14.1 | 19.1 |
| | 2 | 1.2 | 3.5 | 5.2 | 7.5 | 10.4 | 14.1 | 18.9 |
| | 3 | 1.2 | 3.5 | 5.2 | 7.4 | 10.3 | 14.1 | 19.0 |
| Residential | 1 | 1.0 | 3.2 | 5.0 | 7.2 | 10.2 | 14.0 | 19.1 |
| | 2 | 1.0 | 3.2 | 4.9 | 7.0 | 9.8 | 13.3 | 18.0 |
| | 3 | 1.0 | 3.2 | 4.9 | 7.1 | 10.0 | 13.7 | 18.5 |
| Transport | 1 | 0.01 | 0.03 | 0.05 | 0.08 | 0.1 | 0.2 | 0.3 |
| | 2 | 0.01 | 0.04 | 0.06 | 0.09 | 0.1 | 0.2 | 0.2 |
| | 3 | 0.01 | 0.04 | 0.06 | 0.09 | 0.1 | 0.2 | 0.2 |

**Table 10. Existing and predicted the energy demand ($10^3$ GWh) for the city.**

| Sector | Year | | | | | | |
|---|---|---|---|---|---|---|---|
| | 2016 | 2025 | 2030 | 2035 | 2040 | 2045 | 2050 |
| Commercial | 0.9 | 2.7 | 4.1 | 5.9 | 8.3 | 11.3 | 15.2 |
| Industrial | 1.4 | 3.9 | 5.8 | 8.2 | 11.2 | 15.2 | 20.4 |
| Residential | 1.3 | 3.7 | 5.5 | 7.8 | 10.8 | 14.8 | 19.8 |
| Transport | 0.02 | 0.04 | 0.06 | 0.09 | 0.1 | 0.2 | 0.3 |

After the water management intervention, the supply and demand balance index becomes greater and equal to 1. In the 2050 years, the total water supply is 1426 MCM, i.e. 866 MCM more than the reference supply. There should be enough storage capacity, so that excess water from abundant water years be used in the next year. This will be good for water resources and sustainable development.

## 3.5. Analysis of energy supply intervention

The existing and planned energy supply by energy utility to Adds Ababa city in 2030 and 2050 will unmet the future demand. Therefore, alternative sustainable energy supply will be required to improve the unmet energy demand of Addis Ababa city, by considering the energy conservation or demand management option. The other measure is the interdependence of water and energy or water-energy nexus, this approach can also improve the future energy supply. Hence as water is conserved; the energy use for water is also conserved. In 2050, around 54% of water is saved and due to this, about 12–30% of energy is also conserved [35]. The future energy supply scenarios by energy conservation and demand management alternatives are indicated in Table 13.

The total energy saving by ECDM considering efficient lighting in commercial, residential and industrial sector, efficient machine motor in industry, efficient cook stove and mittad in household, as well as loss reduction in distribution system will save 18.26 PJ and 101 PJ in 2030 and 2050 respectively. By the ECDM scenario, the unmet energy demand will grow to 10331 GWh by 2030 and 27414 GWh by 2050; this value is about 63 and 166% decrease respectively when compared with the existing and planned energy supply. Energy produced by solar PV is estimated based on power (103 MW) of solar PV and daily bright sunshine hour (7.3 hr/

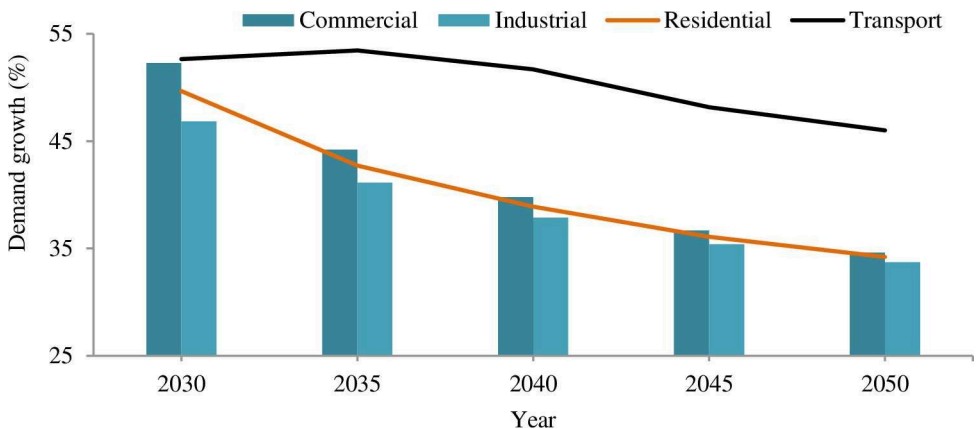

**Fig 12. Energy demand growth rate.** Within a five year interval (2025–2050), the energy growth rate for these sectors is 33–53%. The growth rate for residential, commercial, industrial and street-lighting were 34–50%, 34–42%, 33–46% and 46–53% respectively.

**Table 11. Percent (%) of water saving by WCDM measures.**

| Measure | Year |
| --- | --- |
| | 2050 |
| Replacing older inefficient toilet fixtures | 0.82 |
| Water loss reduction | 4.03 |
| Conservation and improving billing system | 43.2 |
| Distributing low flow showerhead and faucets | 0.85 |
| Installing low flow toilets and urinals | 4.64 |
| Cloth washer rebate | 0.67 |
| Total | 54 |

day), then the annual harvestable energy potential of Addis Ababa city becomes around 275 GWh.

According to the results, about 30 and 50% of energy saving will reflect the demand in 2030 and 2050 respectively. The total energy supply in 2030 and 2050 due to the energy management intervention will become 14186 and 38450 GWh respectively. The energy supply-demand balance index after the intervention for the respective years will be around 0.92 and 0.70.

## 4. Conclusions

This paper conducted sustainable WE supply and demand analysis for Addis Ababa city by considering the socioeconomic, technology factors, ECDM and WCDM strategies. A regression model using the WEKA tool was used to predict these demands. The drivers used in WE consumption estimation were considered based on appropriate goodness of fit parameters. The average scenario ($X_1$ scenario and $X_3$ scenario), gave a better estimation of energy consumption. WE demand was predicted considering the distribution loss system as well as WE consumption. The average $X_1$ and $X_3$ drivers are used to predict energy consumption for residential, commercial, and industrial sectors, while the $X_1$ and $X_3$ drivers are used for the street-lighting sector.

The energy demand was estimated to be 15188 GWh for the commercial sector; 20364 GWh for the industrial sector; 19801 GWh for the residential sector and 292 GWh for the street-lighting sector by 2050. Consequently, for 2050 year, the water demand was estimated to be 227 MCM for the commercial, 818 MCM for the residential and 154 MCM for the industrial sector.

The used model results are consistent with the actual WE demand data, which means that the model has achieved satisfactory levels of accuracy and meets the actual predicting requirements. Based on the predicted WE demand, different options of supply intervention were developed to improve the future unmet demand. In 2050, for the reference WE supply (existing and planned), there will be a high amount of unmet WE demand, which is about 638 MCM and 45702 GWh for water and energy respectively. In the case of water, due to water

**Table 12. Annual total water supply-balance index.**

| Evaluation | Year | | | | | | |
| --- | --- | --- | --- | --- | --- | --- | --- |
| | 2016 | 2025 | 2030 | 2035 | 2040 | 2045 | 2050 |
| Total supply (MCM) | 393 | 809 | 823 | 838 | 896 | 1111 | 1426 |
| Total demand (MCM) | 254 | 355 | 431 | 561 | 716 | 922 | 1198 |
| Index | 2 | 2 | 2 | 1 | 1 | 1 | 1 |

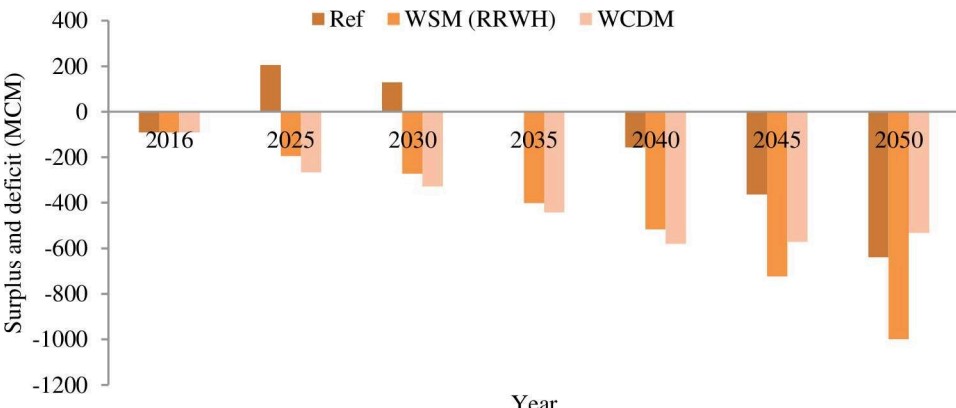

**Fig 13. Water supply-demand gap.** The water supply increases in 2030 and 250 by 823 and 1426 MCM respectively through the water management intervention. The new water supply and balance index can be seen in Table 12.

**Table 13. Estimated energy saving in Giga Watt hour (GWh) using ECDM measure.**

| Measures | Year | | | | | | | |
|---|---|---|---|---|---|---|---|---|
| | 2016 | 2020 | 2025 | 2030 | 2035 | 2040 | 2045 | 2050 |
| Energy efficient lighting (CFL) | 236.11 | 522.22 | 1138.89 | 2272.22 | 4313.89 | 7980.56 | 10972.22 | 14722.22 |
| Efficient energy machine (induction) | 152.78 | 327.78 | 691.67 | 1352.78 | 2544.44 | 4675 | 6394.44 | 8552.78 |
| Efficient mittad (induction) | 66.67 | 100 | 172.22 | 297.22 | 508.33 | 872.22 | 1136.11 | 1463.89 |
| Energy loss reduction technology | 27.78 | 366.67 | 497.22 | 647.22 | 841.67 | 944.44 | 1033.33 | 1113.89 |
| Improved stove | 122.22 | 177.78 | 297.22 | 502.78 | 852.78 | 1447.22 | 1863.89 | 2377.78 |
| Total saving | 602.78 | 1491.67 | 2800 | 5072.22 | 9063.89 | 15916.67 | 21400 | 28230.56 |

conservation and demand management measures, the surplus water will be achieved in 2030 and 2050. Similarly, the energy efficiency measure will highly improve the energy supply of the city with renewable solar energy. The water-energy balance index after the intervention will become greater around 0.7 and 1 for energy and water respectively. Therefore, in 2030 and 2050, from the water-energy supply intervention analysis, the unmet water demand will be met, whereas the energy demand is improved. The water and energy management, from the supply and demand side, basically from the demand side will sustainably improve the future urban water-supply challenge. Therefore based on the estimated WE demand, the city government shall be concentrate on focusing the water and energy management to sustainably meet the demand, rather than developing a new capacity of water and energy supply.

## Supporting information

**S1 Table. CSA and BoFED data source, socio-economic trend for Addis Ababa city, which include population, Gross Domestic Product (GDP) and Per Capita Income (PCI) for the period 2005 to 2016.**
(DOCX)

**S2 Table. AAWSA data source, disaggregated water consumption by end-users in million cubic meters for Addis Ababa city from 2016 to 2020.**
(DOCX)

**S3 Table. EEU data source, electric energy consumption by sectors from 2015 to 2019.**
(DOCX)

**S1 Appendix.**
(DOCX)

## Acknowledgments

This study would like to thanks Bureau of Finance and Economic Development (BoFED), Addis Ababa Water and Sewerage Authority (AAWSA), Central Statistical Agency (CSA) and Ethiopian Electric Utility (EEU) for providing data.

## Author Contributions

**Conceptualization:** Bedassa Dessalegn Kitessa, Semu Moges Ayalew, Geremew Sahilu Gebrie, Solomon T/mariam Teferi.

**Data curation:** Bedassa Dessalegn Kitessa, Geremew Sahilu Gebrie.

**Formal analysis:** Bedassa Dessalegn Kitessa.

**Methodology:** Geremew Sahilu Gebrie.

**Supervision:** Semu Moges Ayalew.

**Validation:** Semu Moges Ayalew, Geremew Sahilu Gebrie.

**Visualization:** Semu Moges Ayalew, Geremew Sahilu Gebrie, Solomon T/mariam Teferi.

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
