## [Decision Letter · Decision Letter 0]

23 Mar 2021

Assessing the Supply for a Basic Urban Service Demand – With a Focus on Water-Energy Management in Addis Ababa City

PONE-D-21-00009

Dear Dr. Kitessa,

We’re pleased to inform you that your manuscript has been judged scientifically suitable for publication and will be formally accepted for publication once it meets all outstanding technical requirements.

This is a very interesting paper. The relationship between water and energy in urban development is always a topic of interest. The calculation process of this paper is very detailed and clear, therefore, I believe it is worthy of publication. If some suggestions are made, my suggestions are:

First, the introduction section should be integrated to outline the scientific problem a little more clearly and to explain the contribution of this study.

Second, the discussion section should be deepened to increase the depth and breadth of the discussion. If possible, the discussion should be a separate chapter.

Third, many of the paragraphs are one or two sentences long, and it is recommended that they be appropriately combined to give a clearer hierarchical structure to the paragraphs.

Within one week, you’ll receive an e-mail detailing any additionally required amendments. When these, and the requests above have been addressed, you’ll receive a formal acceptance letter and your manuscript will be scheduled for publication.

Kind regards,

Bing Xue, Ph.D.

Academic Editor

PLOS ONE

1. Please amend your Data Availability Statement and Methods section to include links or contact information for where the raw data used in this study can was accessed.

2. Thank you for stating the following financial disclosure: 'No'

Please provide an amended Funding Statement that declares *all* the funding or sources of support received during this specific study (whether external or internal to your organization) as detailed online in our guide for authors at http://journals.plos.org/plosone/s/submit-now

Please state what role the funders took in the study.  If any authors received a salary from any of your funders, please state which authors and which funder. If the funders had no role, please state: "The funders had no role in study design, data collection and analysis, decision to publish, or preparation of the manuscript."

c. Please send your amended statements by return email; we will change the online submission form on your behalf.

Reviewers' comments:

Reviewer's Responses to Questions

**Comments to the Author**

1. Is the manuscript technically sound, and do the data support the conclusions?

Reviewer #1: Yes

2. Has the statistical analysis been performed appropriately and rigorously? 

Reviewer #1: N/A

3. Have the authors made all data underlying the findings in their manuscript fully available?

Reviewer #1: Yes

4. Is the manuscript presented in an intelligible fashion and written in standard English?

Reviewer #1: Yes

5. Review Comments to the Author

Reviewer #1: Review Comments to the Author

Please use the space provided to explain your answers to the questions above. You may also include additional comments for the author, including concerns about dual publication, research ethics, or publication ethics. (Please upload your review as an attachment if it exceeds 20,000 characters) (Limit 200 to 20000 Characters)

"have no comments"

6. PLOS authors have the option to publish the peer review history of their article (what does this mean?). If published, this will include your full peer review and any attached files.

Reviewer #1: **Yes: **Tadesse Weyuma

---

## [Editor Report · Acceptance letter]

19 Aug 2021

PONE-D-21-00009 

Assessing the Supply for a Basic Urban Service Demand-With a Focus on Water-Energy Management in Addis Ababa City 

Dear Dr. Kitessa:

I'm pleased to inform you that your manuscript has been deemed suitable for publication in PLOS ONE. Congratulations! Your manuscript is now with our production department. 

Kind regards, 

on behalf of

Professor Bing Xue 

Academic Editor

PLOS ONE